# Transcriptomics Analysis of Lens from Patients with Posterior Subcapsular Congenital Cataract

**DOI:** 10.3390/genes12121904

**Published:** 2021-11-27

**Authors:** Xiaolei Lin, Hongzhe Li, Tianke Yang, Xin Liu, Fan Fan, Xiyue Zhou, Yi Luo

**Affiliations:** Department of Ophthalmology, Eye & ENT Hospital, Shanghai Medical College, Fudan University, Shanghai 200031, China; 19111260013@fudan.edu.cn (X.L.); 14301050163@fudan.edu.cn (H.L.); 20111260011@fudan.edu.cn (T.Y.); xin.liu@fdeent.org (X.L.); fanfan432121@hotmail.com (F.F.); 13301050257@fudan.edu.cn (X.Z.)

**Keywords:** posterior subcapsular congenital cataract, transcriptomics analysis, lens epithelial cells, lens fibre cells

## Abstract

To gain insight into the aetiology of posterior subcapsular congenital cataract from the perspective of transcriptional changes, we conducted an mRNA sequencing analysis of the lenses in posterior subcapsular congenital cataract patients and in normal children. There were 1533 differentially expressed genes from 19,072 genes in the lens epithelial cells of the posterior subcapsular congenital cataract patients compared to in the normal controls at a cut-off criteria of |log2 fold change| of >1 and a *p*-value of <0.05, including 847 downregulated genes and 686 upregulated genes. To further narrow down the DEGs, we utilised the stricter criteria of |log2 fold change| of >1 and an FDR value of <0.05, and we identified 551 DEGs, including 97 upregulated genes and 454 downregulated genes. This study also identified 1263 differentially expressed genes of the 18,755 genes in lens cortex and nuclear fibres, including 646 downregulated genes and 617 upregulated genes. The downregulated genes in epithelial cells were significantly enriched in the structural constituent of lenses, lens development and lens fibre cell differentiation. After filtering the DEGs using the databases iSyTE and Cat-Map, several high-priority candidate genes related to posterior subcapsular congenital cataract such as *GRIFIN*, *HTRA1* and *DAPL1* were identified. The findings of our study may provide a deeper understanding of the mechanisms of posterior subcapsular congenital cataract and help in the prevention and treatment of this disease.

## 1. Introduction

Congenital cataract is the crystalline lens opacification that is present at birth or early childhood [1]. It is derived from changes during embryonic development and leads to significant visual impairment [2]. Congenital cataract is responsible for 5–20% of preventable blindness in children, and the prevalence of congenital cataract is the highest in Asia (7.43/10,000) compared with in other regions of the world [3]. Preventing and treating congenital cataract without delay is paramount in alleviating the sufferings of patients and their families and societies.

Previous studies have found that congenital cataracts can be caused by inherited gene mutations or intrauterine diseases, such as infectious, metabolic or drug-induced complications during pregnancy. However, as the pathogenic mechanisms of congenital cataract remain unclear, about 50% of bilateral congenital cataracts and almost all unilateral congenital cataracts have no identified cause and are diagnosed as idiopathic congenital cataract [3,4]. The prevention and early treatment of these idiopathic congenital cataracts are difficult.

The gene expression information of the specific tissue is critical to detect candidate genes associated with developmental defects [5,6,7]. High-throughput mRNA sequencing analysis can represent transcripts of all the transcribed genes of a given tissue, which is crucial to study the mechanisms of congenital cataract. Comparing the expression levels of genes in different congenital cataract patients with normal controls helps in the detection of transcriptional changes and clarifies the pathogenesis of congenital cataract. However, research on the transcripts of crystalline lens in congenital cataract patients is rarely addressed.

Previous studies have suggested that the morphology of cataract may be a predictor of visual prognosis [8]. There is great variability in visual outcomes in different congenital cataract patients after surgery. Patients with a greater density of cataract in the central visual axis at a younger age often have more severe stimulus deprivation amblyopia, which results in poorer vision [8]. While the pathogenesis of different types of congenital cataracts is still uncertain, the posterior subcapsular congenital cataract is one of the most common types and has a great effect on visual acuity [3].

In this study, we attempt to explain the pathogenesis of posterior subcapsular congenital cataract from the perspective of transcriptional changes. The transcriptome approach is used in the present study to compare the mRNA expression in patients having posterior subcapsular congenital cataract with that in normal children. The findings of our study may provide researchers with a deeper understanding of the mechanisms of posterior subcapsular congenital cataract and help in the prevention and treatment of this disease.

## 2. Materials and Methods

### 2.1. Subjects

This study was conducted adhering to the tenets of the Helsinki Declaration and was approved by the Investigational Review Board of the Eye and ENT Hospital of Fudan University (protocol code: 2020076; date of approval: August 2020), Shanghai, China. Informed consent was obtained from the guardians of participants. Patients with bilateral or unilateral posterior subcapsular congenital cataracts under 10 years of age were included in the study. For children with posterior subcapsular congenital cataract, the exclusion criteria were as follows: ocular inflammation, persistent fetal vasculature (PFV), persistent hyperplastic primary vitreous (PHPV), pre-existing posterior capsule defect, combined other developmental disorder or a history of ocular surgery or trauma. The normal controls included transparent lenses from children younger than 10 years of age.

Anterior lens capsules and the cortical and nuclear fibres of 6 congenital cataract patients were obtained during their cataract surgery by manual continuous curvilinear capsulorhexis and cataract aspiration by the same surgeon (Y.L.). The materials of 8 normal controls were obtained from the Eye Bank of the Eye and ENT Hospital of Fudan University within 24 h post-mortem. The anterior lens capsules were gently torn using micro-forceps after being cut with a blunt-tipped Vannas scissors; the cortical and nuclear fibres of lenses were also collected. 

To further verify the results of RNA sequencing (RNA-seq), another 24 posterior subcapsular congenital cataract patients (age: 4–110 months) and 17 normal controls (age: 19 days–84 months) were enrolled in the study, and the materials were obtained using the same method described above.

### 2.2. RNA Extracting, Library Construction and RNA-seq

Total RNA was extracted from the anterior lens capsules using the Quick-RNA™ Microprep Kit (Zymo Research, Orange, CA, USA), in accordance with the manufacturer’s protocol. The concentration of total RNA was measured with Qubit 4.0 (Thermo Fisher Scientific, Waltham, MA, USA). The RNA quality was analysed using agarose gel electrophoresis. The amplification products of full-length complementary DNA (cDNA) were obtained using the Single Cell Full Length mRNA-Amplification Kit (N712-01, Vazyme, Nanjing, China). The mRNA was purified using Oligo (dT) VN Primer (Vazyme, Nanjing, China). Complementary DNA was produced via reverse transcription. The full-length cDNA was amplified by PCR under the following conditions: 98 °C for 1 min, 14–24 cycles of 98 °C for 10 s, 65 °C for 15 s and 72 °C for 6 min; followed by 72 °C for 5 min; 4 °C hold. The cDNA quality was analysed using the Agilent 2100 Bioanalyzer (Agilent Technologies, Santa Clara, CA, USA).

The TruePrep^®^ DNA Library Prep Kit v2 for Illumina (TD502, Vazyme, Nanjing, China) was used to construct the cDNA libraries. In brief, 5 μL TTE (Vazyme, Nanjing, China) mix V50, 4 μL 5 × TTBL, 15 ng cDNA, and ddH_2_O up to 20 ul were mixed and blended. The cDNA was fragmented following PCR conditions: 55 °C for 10 min; 10 °C hold. The reaction was stopped by incubating with 5 μL 5 × TS for 5 min at room temperature. PCR was performed after adding 25 μL cDNA, 5 μL 5 × TAB, 5 μL N5XX primer, 5 μL N7XX primer, 4 μL ddH_2_O (Vazyme, Nanjing, China), and 1 μL TAE (Vazyme, Nanjing, China) under the following conditions: 72 °C for 3 min; 98 °C for 30 s; and 14 cycles of 98 °C for 15 s, 60 °C for 30 s and 72 °C for 3 min; followed by 72 °C 5 min; 4 °C hold. The quality of the PCR products was analysed using 2% agarose gel electrophoresis (Thermo Fisher Scientific, Waltham, MA, USA), and the libraries were purified using DNA Clean Beads (Vazyme, Nanjing, China) and were sequenced using the Illumina Nova6000 platform (Illumina, San Diego, CA, USA).

### 2.3. Gene Expression Data Analysis

All reads were mapped to GRCh38 using the STAR software (version 2.5.3a). For all samples, String Tie software (version 1.3.1c) was used to count the original sequence counts of known genes, and the expression levels of known genes were quantile-normalised using the fragments per kilobase of transcript per million fragments mapped (FPKM) value for further analysis. Differentially expressed genes (DEG) between different sample groups were screened using the DESeq2 software (version 1.16.1).

Functional annotation and clustering analysis were performed on all DEGs by gene ontology (GO) and Kyoto Encyclopaedia of Genes and Genomes (KEGG) analysis. The GO enrichment analysis was used to identify the main biological functions of DEGs via the TopGO software (version 2.42.0), and the KEGG pathway database helped to confirm the information on how molecules or genes are networked (http://www.genome.jp, accessed on 23 January 2021). Dynamic RNA expression data for DEGs in mouse lens during embryonic development were obtained from the iSyTE database as described previously [5,6].

Protein–protein interaction (PPI) networks for differentially expressed proteins were also analysed using the Search Tool for the Retrieval of Interacting Genes (STRING) database (http://string-db.org, accessed on 21 January 2021) [9].

### 2.4. Real-time Quantitative PCR (RT-qPCR)

The expression of DEGs identified by RNA-seq analysis was validated using quantitative reverse transcription PCR (RT-qPCR). Total RNA was isolated from the anterior lens capsules and cortex using TRIzol reagent (Invitrogen, Waltham, MA, USA). It was then used to synthesise cDNA using PrimeScript™ RT reagent Kit with gDNA Eraser (TaKaRa, Tokyo, Japan), followed by RT-PCR using SYBR^®^ Green Premix Pro Taq HS qPCR Kit (AG11705, Accurate Biotechnology, Changsha, China) on Bio-Rad CFX96 Touch Real-Time PCR Detection System and software v4.1 (Bio-Rad, Hercules, CA, USA). The gene-specific primers for RT-qPCR are shown in Table 1. The results were analysed using the 2^−ΔΔCt^ method as described previously [10].

### 2.5. Immunohistochemical Staining

After fixation overnight, the whole lens of the normal controls from the Eye Bank were embedded in paraffin and then cut into 3 μm-thick sections. The sections were deparaffinised and rehydrated; subsequently, the sections were treated with sodium citrate buffer (pH 6.0) for antigen retrieval and incubated in 3% hydrogen peroxide for 15 min at room temperature to block endogenous peroxidase. After blocking, the sections were incubated in *HTRA1* primary antibodies (1:200) at 4 °C overnight. Immunodetection was performed using DAB substrate liquid (Thermo Fisher Scientific, Waltham, MA, USA). Then, the sections were counterstained with haematoxylin and observed with the same microscope (Leica, Wetzlar, Germany).

### 2.6. Statistical Analysis

Differential expression statistics were assessed using DESeq2, and the adjusted *p*-values which were the false discovery rates (FDRs) were obtained using the Benjamini–Hochberg multiple test correction method. The genes with |log2 fold change|of >1 and a *p*-value of <0.05 were defined as a DEG between the two groups. The GO terms and the KEGG pathway with *p*-values less than 0.05 were considered to be significantly enriched. Statistical analyses for age and RT-qPCR results were performed using SPSS 22.0 (IBM, Armonk, NY, USA). The Kolmogorov–Smirnov test was used for normality distribution. An independent sample *t* test or Mann–Whitney U test was performed to analyse the difference between the posterior subcapsular congenital cataract patients and the normal control group. *p*-values less than 0.05 were considered statistically significant.

## 3. Results

A total of 14 children were included in the study for RNA-seq, of which 6 patients had posterior subcapsular congenital cataracts (4 bilateral and 2 unilateral; age: 51–108 months) and 8 normal controls (age: 2–96 months). All the congenital cataract patients were non-syndromic and had a negative family history. An average of 110,375,304 (79,150,688–167,149,114) raw reads was yielded per sample. After the data were filtered, each sample generated an average of 53,416,765 (23,475,256–113,734,102) reads (Table 2). The percentage of the aligned reads to GRCh38 ranged from 79.1% to 97.9%. The results showed good sequence quality.

### 3.1. Differential Expression and Functional Enrichment Analysis in the Anterior Lens Epithelial Cells

In total, 19,072 genes were identified in the anterior lens capsule samples. The results showed 1533 DEGs at the cut-off criteria of |log2 fold change| of >1 and a *p*-value of <0.05. Using the cut-off criteria of |log2 fold change| of >1 and an FDR value of <0.05, we identified 551 DEGs, including 97 upregulated genes and 454 downregulated genes. The volcano plots and the heat map of DEGs are shown in Figure 1a,b. The top five most upregulated and downregulated DEGs are presented in Table 3.

The identified DEGs were significantly associated with the structural constituents of eye lens (such as *BFSP1*, *CRYBA1* and *CRYBA2*), lens development in the camera-type eye (such as *MIP*, *PROX1* and *WNT7A*), response to cadmium ion (such as *SORD*, *FOS* and *GPI*), cytoplasm (such as *ABRAXAS2, ACAD11 and ACTA1*), structural molecule activity (such as *MIP*, *JAG1* and *LIM2*) and lens fibre cell differentiation (such as *WNT7B*, *TDRD7* and *TMOD1*) (Figure 2a). In addition, KEGG pathways were enriched for these DEGs, including steroid biosynthesis (such as *FDFT1*, *CYP51A1* and *SQLE*), p53 signalling pathways (such as *CCNG1*, *CDKN1A* and *CCND2*), nitrogen metabolisms (such as *CA2*, *CA4* and *CA14*), mineral absorption (such as *MT1F*, *MT1H* and *HMOX1*) and glycine, serine and threonine metabolisms (such as *PHGDH*, *MAOA* and *BPGM*) (Figure 2b).

### 3.2. Differential Expression and Functional Enrichment Analysis in the Fibre Cells

The fibre cells expressed a total of 18,755 genes. Using the threshold of a *p*-value of ≤0.05 and |log2 fold change| of >1, 1263 DEGs were identified from the cortical and nuclear fibres, including 646 downregulated genes and 617 upregulated genes. The volcano plots and the heat map were shown in Figure 1c,d.

The GO annotation indicated that the DEGs were significantly related to the regulation of wound healing (such as *APOE* and *CCL2*), the regulation of response to wounding (such as *GJA1*, *WNT4* and *APOE*), extracellular matrix (ECM) organisations (such as *COL11A2*, *SULF1* and *SFRP2*), extracellular structure organisations (such as *APOE*, *COL11A2* and *SFRP2*) and ECMs (such as *BMP7*, *COL11A1* and *COL11A2*) (Figure 2c). Meanwhile, the KEGG pathway enrichment analysis showed that the DEGs were enriched in Rap1 signalling pathways (such as *EFNA3*, *MAPK12* and *GRIN2B*), glycolysis/gluconeogenesis (such as *FBP1*, *PGAM2* and *ALDH3A1*) and glycine, serine and threonine metabolisms (such as *PSAT1*, *CBSL* and *BPGM*) (Figure 2d).

### 3.3. PPI Network/Strings Network

The top 50 PPI networks for DEGs were constructed (Appendix A). In the PPI network, more nodes indicate higher connectivity.

### 3.4. RT-qPCR Validates the Results of RNA-seq

The top five most upregulated and downregulated DEGs in the anterior lens epithelial cells were selected to validate the results of RNA-seq. The results showed that the RNA expression of these genes obtained by RT-qPCR was in concordance with the results of RNA-seq (Table 3; Figure 3a). 

Nine DEGs in the fibre cells were also selected for validation via RT-qPCR. The changes of five downregulated genes including *APOE*, *TSPAN12*, *PRSS35*, *ASS1* and *SULF1* and three upregulated genes including *PRAP1, RAMP1* and *TMEM54* were consistent with the data obtained from RNA-seq (Table 4; Figure 3b).

### 3.5. Lens-Associated Differential Gene Expression to Identify High-Priority Candidate Genes

To identify the candidate genes from the DEGs obtained via RNA-seq analysis that may be associated with the pathology of posterior congenital cataract, we next analysed the DEGs using the database resource of iSyTE2.0 [5], which represented the gene expression and lens enrichment data during the normal lens development. Previous studies have suggested that the database iSyTE could be a cataract gene prioritization resource [10] and 95% of known genes associated with cataract or lens defects are lens-enriched at one or more key embryonic stages [6]. 

Amongst the microarray database on the Affymetrix 430 2.0 platform in iSyTE, 62 of the DEGs of the anterior lens epithelial cells were ranked within the 528 top lens-enriched genes from all lens stages. From the 62 DEGs, 21 were known genes associated with cataract according to the Cat-Map database [11]. The other 41 genes were candidate cataract-associated genes. After ranking the 41 genes in iSyTE2.0 according to the expression from E10.5 through P56, *GRIFIN*, *DAPL1*, *LGALS3*, *GPR160*, *MBOAT1*, *HTRA1*, *SLC7A2*, *PIP5KL1*, *CAPRIN2* and *ARSI* were suggested as the top 10 high-priority candidate genes. In these genes, *HTRA1* was the most downregulated DEG in the result of RNA-seq, which was selected for further validation.

According to the data obtained through the transcriptomics profile of mouse embryonic lens development based on RNA-seq [6], 52 DEGs in the anterior lens epithelial cells were found to be highly enriched in the lens at one or more stages. These genes included 52 genes enriched in E10.5, 42 genes enriched in E12.5, 37 genes enriched in E14.5 and 38 genes enriched in E16.5. Amongst the 52 DEGs, 16 were known genes associated with cataract according to the database Cat-Map [11], including *ADAMTSL4*, *APOE*, *BFSP1*, *CRYBA1*, *CRYBA2*, *CRYBA4*, *CRYBB1*, *CRYBB3*, *CRYGA*, *CRYGB*, *CRYGC*, *CRYGD*, *EFNA5*, *GJA3*, *MIP* and *PROX1*. The other 36 genes were candidate cataract-associated genes. After ranking the 36 genes in iSyTE2.0 according to the expression from E10.5 through E16.5, *MBOAT1*, *NGEF*, *MED12L*, *DAPL1* and *JAG1* were suggested as the top 5 high-priority candidate genes. 

The same method was used to analyse the DEGs in the fibre cells. A total of 49 DEGs of the lens fibre cells were ranked within the 528 top lens-enriched genes from all lens stages. After filtering the known genes associated with cataract, 39 genes were identified as candidate cataract-associated genes. After ranking the transcriptomics profile based on RNA-seq [6], 209 DEGs in the lens fibre cells were found to be highly enriched in the lens at one or more stages. Amongst these genes, 18 were known cataract-associated genes. The other 191 genes were candidate cataract-associated genes. According to the results of the 2 filtering methods, *XIST*, *GRIFIN*, *HTRA3* and *CLU* were all high-priority candidate genes. 

### 3.6. A Protein Marker of Cell Growth and Epithelial Cell Proliferation Confirms RNA-seq Transcriptome Findings

To confirm the observations from the RNA-seq analysis, the expression of *HTRA1* was verified at the RNA level and the protein level. Anterior lens capsules from another 24 posterior subcapsular congenital cataract patients and 12 normal controls were used for RT-qPCR (Appendix A). The age did not differ significantly between the congenital cataract patients and normal controls (*p*-value > 0.05). The results showed that the mRNA expression of *HTRA1* was significantly downregulated in posterior congenital cataract patients compared with in normal controls, and the mean fold change was 158.62 (*p*-value < 0.001). Immunohistochemistry indicated that the *HTRA1* protein was located in the epithelial cells and outer fibre cells (Figure 4). In addition, the protein expression of *HTRA1* seemed to be reduced with age (Figure 4a–e).

## 4. Discussion

Congenital cataract is a crucial and preventable cause of childhood blindness. The prevalence of posterior subcapsular congenital cataract is 26.8% worldwide, and it is one of the most common types of congenital cataracts [3]. The pathogenesis of posterior subcapsular congenital cataract is still largely unknown. The RNA-seq can acquire the total number and type of transcripts and identify different transcripts within different tissues. To analyse type-specific gene expression patterns, the anterior lens capsules, cortex and nuclear fibres in posterior subcapsular congenital cataract patients and normal controls were collected for RNA-seq. The results of this study can provide some important insights into the molecular pathology of posterior subcapsular congenital cataracts. 

The analysis workflow in this study was conducted as follows: data standardisation and quality control to identify DEGs, functional annotation and clustering analysis followed by filtering of the DEGs using the databases iSyTE and Cat-Map [12]. The database iSyTE allows the visualisation and analysis of dynamic gene expression in lens development and disease, helps to identify new cataract-linked genes [5] and provides an estimate of the expression of lens enrichment. Previous studies have used the database to identify new genes associated with cataracts [5,12,13,14]. 

We found that the expression levels of more than 1533 of the 19,072 genes in the lens epithelial cells of the posterior subcapsular congenital cataract patients differed more than twofold compared to those of the normal controls. Of these, 686 genes were upregulated, and 847 genes were downregulated in the posterior subcapsular congenital cataract patients using the cut-off criteria of |log2 fold change| of >1 and an FDR value of <0.05. The functional cluster analysis of the identified genes showed that a variety of biological pathways changed significantly after posterior subcapsular congenital cataract formation, including lens cell development, cell differentiation and growth and cadmium ion. Choudhary et al. found that six weeks of CdCl_2_ administration results in a considerable progress of cataract formation, and the antioxidants for lenses, such as CAT, SOD, GPx and GSH, are significantly reduced [15]. Heavy metal ions might play a potential role in cataract pathology [15,16,17]. It is quite different from the results of age-related cataract research, suggesting that the mechanism of congenital cataract is different from that of age-related cataract, which is in accordance with previous studies [18]. Oligonucleotide microarrays were used in the study of Hawse et al. to compare the global gene expression profiles of epithelial cells between age-related cataracts and clear lenses [19]. Their results showed that functional clustering is significantly altered in oxidative stress, protein synthesis and ion transport pathways [19]. After analysing the upregulated and downregulated DEGs separately, we found that the downregulated genes were significantly enriched in the structural constituent of lens, lens development and lens fibre cell differentiation. In the top 10 downregulated genes, 9 of them were gene-encoding lens crystalline. The results indicated that the biological process of posterior subcapsular congenital cataract formation may be caused by damaging the lens structural proteins, affecting the cell development and differentiation. Interestingly, the results showed that there were five significantly upregulated genes, including *HBA2*, *HBA1*, *HBB*, *HBG1* and *HBG2*, which encoded different haemoglobin chains and were enriched in haemoglobin complex, oxygen carrier activity and oxygen transport. The reason for these results cannot exclude the probability of residual red blood cells during the surgery. Although we have used phosphate-buffered saline (PBS) to clean the samples for at least three times to avoid the remaining reticulocytes after the samples were taken out from the incision, a small number of red blood cells might remain in the sample. More samples should be enrolled to verify the function and mechanism of high-priority candidate genes in further studies.

This study also identified 1263 DEGs of the 18,755 genes in lens cortex and nuclear fibres, including 646 downregulated genes and 617 upregulated genes. The functional cluster analysis suggested that the downregulated genes are significantly enriched in wound healing, organic acid metabolic processes, oxoacid metabolic processes and carboxylic acid metabolic processes, which are mostly related to acid metabolism, whereas the upregulated genes are significantly enriched in the component of plasma membrane and signalling receptor activity. We can assume that with the acid metabolic process in the lens altered, the oxygen uptake is changed and some signalling pathways are activated.

The DEGs in lens epithelial cells showed that gene-encoding crystalline lenses are the most downregulated genes, including *CRYGC*, *CRYGB*, *CRYGA*, *CRYBB1*, *CRYBA1*, *CRYBA4*, *CRYGD* and *CRYBA2*. Crystallin genes are the main constitutive proteins of lenses that play a crucial role in maintaining the refractive properties and transparency of the lenses. Previous studies have found that about 50% of non-syndromic, inherited, congenital cataracts are caused by mutations in the gene-encoding crystalline lenses [20,21]. These results further confirmed the pivotal role of crystallin genes.

The results also found that galectin-related inter-fibre protein (*GRIFIN*) was a DEG in both lens epithelial cells and fibre cells. The *GRIFIN* is a member of the family of adhesion/growth-regulatory galectins [22,23] and is usually expressed in lens fibre cells of zebrafish, rat and chicken [24,25], and the chicken galectin-related interfibre protein (*C-GRIFIN*) has been identified as a lens-specific protein and acts as a lens-specific galectin [24]. The study by Barton et al. also found that *GRIFIN* is bound with α-crystallin in calf lens [26]. Previous studies have indicated that *GRIFIN* may participate in the regulation of the development of lenses, and as a novel candidate gene in our study, the functions of *GRIFIN* in the development of subcapsular congenital cataract need further study.

The results of the RNA-seq also showed that *HTRA1* is one of the most significant DEGs in epithelial cells. The RT-qPCR result confirmed that *HTRA1* was significantly downregulated in the posterior subcapsular congenital cataract patients compared with in the normal controls. *HTRA1* is a secreted serine-proteases and widely expressed in various types of cells and tissues. It is a lens-enriched gene (significantly highly expressed in the lenses compared to in other tissues) from E16.5 to P28Epi, according to the data from the database iSyTE. Posterior subcapsular congenital cataract is one of the fibrotic cataracts [27]. Transforming growth factor-β (TGF-β) has been suggested as the most important factor in inducing fibrotic cataracts, such as anterior and posterior subcapsular congenital cataracts and posterior capsule opacity. The posterior migration of the lens cells and abnormal terminal differentiation of lens fibre cells have been considered key pathological features of posterior subcapsular congenital cataract. Previous studies have shown that TGF-β1 could induce the rapid lens cell elongation and abnormal accumulation of the ECM, the deposition of α-smooth muscle actin (α-SMA) and cell death due to apoptosis [28,29]. TGF-β signalling also suggests a key event during fibre terminal differentiation [30]. The lens-specific transgenic TGF-β1 mice showed subcapsular plaque formation and posterior subcapsular nucleation and vacuole formation, which resembled the pathological characteristics of posterior subcapsular congenital cataract. It is reported that *HTRA1* interacts with members of the TGF-β family and regulates their signalling pathways [31]. The GST pull-down analysis showed that *HTRA1* protein was bound to all tested TGF-β family proteins and inhibited their functions [32]. The interaction between *HTRA1* and the TGF-β family has been verified in various diseases, including brain development [31], age-related macular degeneration [33,34] and tumours [35]. As a serine protease, *HTRA1* plays a key role in regulating various cellular processes and participates in cell proliferation, migration and apoptosis. Previous studies have found that *HTRA1* has a clear proteolytic capability and a recognised effect in cartilage degeneration, skeletal disorders and cancers [35], and *HTRA1* can bind to TGF-β1 and degrade and regulate the amount of TGF-β1 in vivo and in vitro [31,36]. Some recent studies have shown that *HTRA1* is implicated in ER stress-induced unfolded protein response that plays a key role in the pathogenesis of congenital cataract [13,14,16]. Therefore, we assume that *HTRA1* may modulate progression of posterior subcapsular congenital cataract by interfering with TGF-β signalling which need further validation.

However, the results also have some limitations. The sample size was small, and could not fully represent the expression profiles of all patients. In future studies, we will recruit more patients with congenital cataracts and will recruit various cataract phenotypes to determine the expression differences between them. In addition, the function and the mechanism of high-priority candidate genes should also be verified by further studies.

## 5. Conclusions

The present study provides an evidence of broad differences in gene expression between the lenses in children with posterior subcapsular congenital cataract and the lenses in normal children and identifies novel candidate genes. The gene functional enrichment analysis might deepen our understanding of posterior subcapsular congenital cataract formation and mechanisms. Furthermore, this study provides promising evidence that can inform future research on the pathological mechanisms of posterior subcapsular congenital cataracts and the screening of clinical candidate drugs. These investigations could help prevent and treat this idiopathic cataract.

## Figures and Tables

**Figure 1 genes-12-01904-f001:**
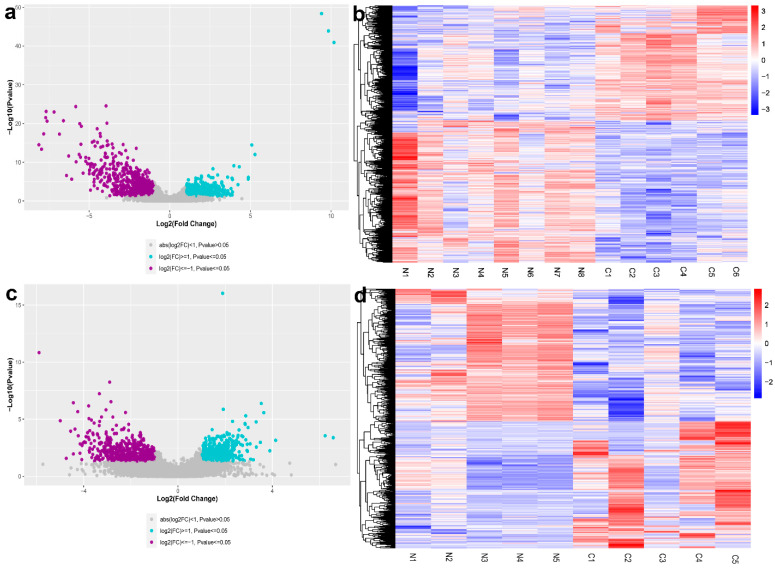
Differentially expressed genes (DEGs) in posterior subcapsular congenital cataract patients compared with in normal control samples. (**a**) Volcano plots of the DEGs in the anterior lens capsule samples. Blue dots indicate upregulated genes, and purple ones represent downregulated genes; (**b**) hierarchical clustering performed on the DEGs in the anterior lens capsule samples; highly expressed genes are shown in pink, and the lower ones are shown in blue; C indicates congenital cataract samples, while N represents normal control samples; (**c**) volcano plots of the DEGs in cortex and nuclear fibres; blue dots indicate upregulated genes, and purple dots represent downregulated genes; (**d**) hierarchical clustering performed on the DEGs in cortex and nuclear fibres.

**Figure 2 genes-12-01904-f002:**
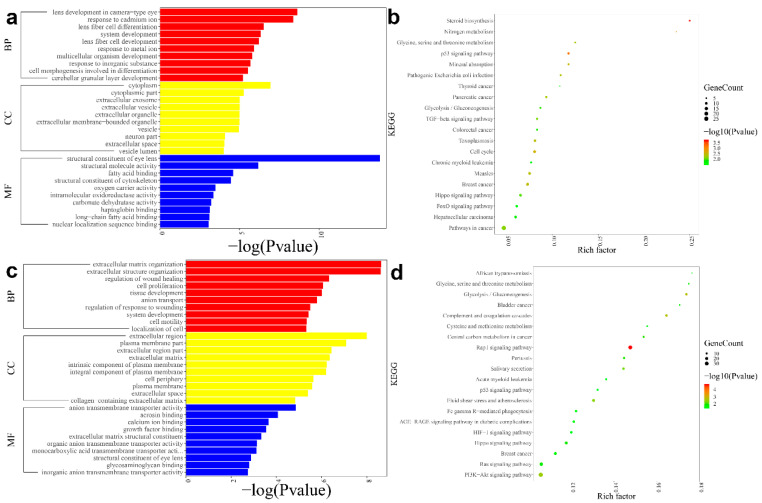
Gene ontology (GO) annotation and Kyoto Encyclopaedia of Genes and Genomes (KEGG) pathway enrichment analysis results for the DEGs of the posterior subcapsular congenital cataract patients compared with those of the normal controls. (**a**) The top 30 GO enrichment of the DEGs in the anterior lens capsule samples; (**b**) the top 20 KEGG enrichment analysis results of the DEGs in the anterior lens capsule samples; (**c**) the top 30 GO enrichment of the DEGs in the cortex and nuclear fibres; (**d**) the top 20 KEGG enrichment analysis results of the DEGs in cortex and nuclear fibres.

**Figure 3 genes-12-01904-f003:**
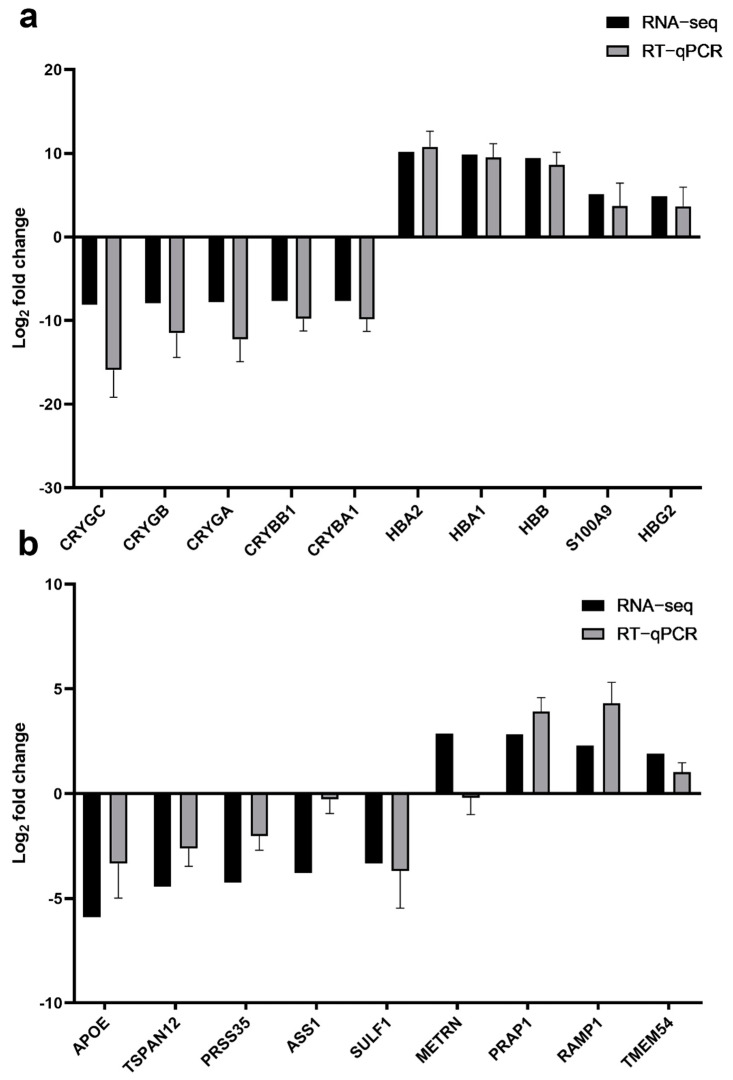
Validation of RNA-sequencing results by RT-qPCR. (**a**) RT-qPCR validation results of the five upregulated genes and the five downregulated genes identified by RNA-sequencing analysis in the anterior lens capsule samples; (**b**) RT-qPCR validation results of the five upregulated genes and the four downregulated genes identified by RNA-sequencing analysis in cortex and nuclear fibres.

**Figure 4 genes-12-01904-f004:**
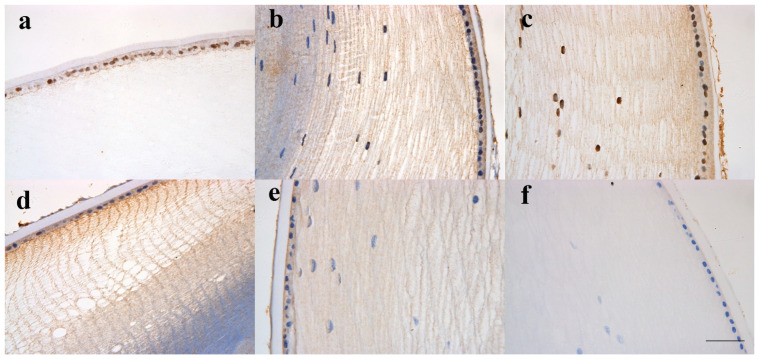
Images of immunohistochemistry of the stained *HTRA1* for transparent lens from donors of different ages. (**a**). Image of a 19-day-old boy; (**b**) image of a 2-month-old boy; (**c**) image of a 3-month-old girl; (**d**) image of a 15-month-old boy; (**e**) image of a 19-year-old girl; (**f**) negative control image of a 2-month-old boy. The results indicated that the *HTRA1* protein was located in the epithelial cells and outer fibre cells. The images are taken at a 20× magnification. Scale bar: 50 μm.

**Table 1 genes-12-01904-t001:** Primers for the real-time quantitative PCR (RT-qPCR) assay.

Target	F/R	Primer Sequence (5′→3′)	Amplicon Size (BP)
CRYGC	Forward	TCCCAACTACCAAGGTCAACA	232
	Reverse	GTGGAGGGAACGGATCTCG	
CRYGB	Forward	CACCTCACTGAAATTCACTCCC	125
	Reverse	CCCCAATCAAGAAACCTCCTGT	
CRYGA	Forward	GGTCCAATCCTGCCGTATAATTC	109
	Reverse	ACAGGCGCAGTCATCAGTG	
CRYBB1	Forward	GTGCTCAAATCTGGCAGACC	92
	Reverse	GGAAGTTGGACTGCTCAAAGG	
CRYBA1	Forward	GGGCAAGAGGATGGAGTTCAC	122
	Reverse	CACAGAAGCTGGTATGCTCATAA	
HBA2	Forward	CAAGACCTACTTCCCGCACTTCG	341
	Reverse	GGGCAGGAGGAACGGCTACC	
HBA1	Forward	TCAACTTCAAGCTCCTAAGCCACTG	110
	Reverse	CACAGAAGCCAGGAACTTGTCCAG	
HBB	Forward	AGGAGAAGTCTGCCGTTACTG	190
	Reverse	CCGAGCACTTTCTTGCCATGA	
S100A9	Forward	CTGTGTGGCTCCTCGGCTTTG	98
	Reverse	TGGTGGAAGGTGTTGATGATGGTC	
HBG2	Forward	GTGGAAGATGCTGGAGGAGAAACC	106
	Reverse	TGATGGCAGAGGCAGAGGACAG	
APOE	Forward	GTTGCTGGTCACATTCCTGG	146
	Reverse	GCAGGTAATCCCAAAAGCGAC	
TSPAN12	Forward	CCAGAGAAGATTCCGTGAAGTG	107
	Reverse	GTCCCTCATCCAAGCAGAAAC	
PRSS35	Forward	CATCGAATGCCAGAAAGAACTCC	135
	Reverse	GTCGGCTCAAGAACCAAATCT	
ASS1	Forward	TCCGTGGTTCTGGCCTACA	126
	Reverse	GGCTTCCTCGAAGTCTTCCTT	
SULF1	Forward	GATCCCCGAGGTTCAGAGGA	178
	Reverse	GGTGTAGTCACAAAGGCATTGA	
METRN	Forward	CACAGACACCGCCAGGCAAG	103
	Reverse	CCACTCCTCTCCCGTCCACAC	
PRAP1	Forward	ACAGCCTGTACCACCCTCC	82
	Reverse	AGCACCTGGTGATTTGGCATC	
RAMP1	Forward	CCAGGAGGCTAACTACGGTG	297
	Reverse	GGGACCACGATGAAGGGGTA	
TMEM54	Forward	TGGGCCATGTGAGCTTCATC	176
	Reverse	GAGGTAGCGTGACAACACGAT	
GAPDH	Forward	GGAGCGAGATCCCTCCAAAAT	197
	Reverse	GGCTGTTGTCATACTTCTCATGG	

**Table 2 genes-12-01904-t002:** RNA-seq samples, read metrics and mapped ratios.

Sample	Tissue	Raw Reads	Clean Reads	Clean Read-Pairs	Average Length	Mapped Ratio (%)
PSC patient 1	Capsules	79,150,688	23,475,256	11,737,628	138.3	88.80%
PSC patient 2	Capsules	92,790,046	36,750,674	18,375,337	141.9	93.40%
PSC patient 3	Capsules	102,532,016	47,760,460	23,880,230	142.0	91.70%
PSC patient 4	Capsules	100,012,388	32,018,416	16,009,208	142.4	92.60%
PSC patient 5	Capsules	98,193,322	36,350,310	18,175,155	134.7	79.10%
PSC patient 6	Capsules	100,226,088	40,541,834	20,270,917	134.1	79.90%
NC 1	Capsules	127,355,192	60,141,116	30,070,558	144.6	95.60%
NC 2	Capsules	100,869,818	37,378,548	18,689,274	138.0	86.40%
NC 3	Capsules	111,902,712	47,398,990	23,699,495	141.5	92.20%
NC 4	Capsules	127,358,610	48,182,942	24,091,471	144.2	94%
NC 5	Capsules	120,259,640	52,969,554	26,484,777	134.6	90.60%
NC 6	Capsules	91,405,500	31,710,346	15,855,173	133.6	84.20%
NC 7	Capsules	125,882,600	56,962,562	28,481,281	136.5	93%
NC 8	Capsules	131,778,762	54,214,656	27,107,328	135.5	91.70%
PSC patient 1	Cortex	123,977,234	45,747,380	22,873,690	141.6	93.40%
PSC patient 3	Cortex	91,869,478	25,684,820	12,842,410	137.9	85.90%
PSC patient 4	Cortex	110,104,320	109,339,070	54,669,535	150.0	97.60%
PSC patient 5	Cortex	128,652,528	48,532,586	24,266,293	138.7	89.40%
PSC patient 6	Cortex	116,294,432	47,393,278	23,696,639	136.5	84.80%
NC 1	Cortex	98,977,182	32,642,774	16,321,387	141.8	92.40%
NC 4	Cortex	167,149,114	67,734,538	33,867,269	142.4	95.10%
NC 5	Cortex	94,542,668	93,252,442	46,626,221	148.9	96.90%
NC 6	Cortex	114,900,736	113,734,102	56,867,051	148.8	97.90%
NC 7	Cortex	92,822,214	92,085,706	46,042,853	149.5	97.90%

PSC, posterior subcapsular congenital cataract; NC, normal control.

**Table 3 genes-12-01904-t003:** The top 5 upregulated and downregulated DEGs in the anterior lens capsule samples of the posterior subcapsular congenital cataract patients.

DEG	Symbol	Description	Log2 Fold Change	*p*-Value	False Discovery Rate (FDR) Value
Downregulated	CRYGC	Crystallin gamma C	−8.117	3.04 × 10^−15^	6.00 × 10^−12^
	CRYGB	Crystallin gamma B	−7.952	4.24 × 10^−14^	6.30 × 10^−11^
	CRYGA	Crystallin gamma A	−7.818	4.67 × 10^−18^	1.61 × 10^−14^
	CRYBB1	Crystallin beta B1	−7.695	3.07 × 10^−22^	2.11 × 10^−18^
	CRYBA1	Crystallin beta A1	−7.667	7.63 × 10^−24^	7.00 × 10^−20^
Upregulated	HBA2	Hemoglobin subunit alpha 2	10.182	1.20 × 10^−41^	2.21 × 10^−37^
	HBA1	Hemoglobin subunit alpha 1	9.838	1.27 × 10^−44^	3.48 × 10^−40^
	HBB	Hemoglobin subunit beta	9.414	4.16 × 10^−49^	2.29 × 10^−44^
	S100A9	S100 calcium binding protein A9	5.087	3.54 × 10^−15^	6.71 × 10^−12^
	HBG2	Hemoglobin subunit gamma 2	4.863	2.18 × 10^−6^	4.59 × 10^−4^

**Table 4 genes-12-01904-t004:** The top 5 upregulated and top 4 downregulated DEGs in the lens fibre cells of the posterior subcapsular congenital cataract patients.

DEGs	Symbol	Description	Log2 Fold Change	*p*-Value	FDR Value
Downregulated	APOE	Apolipoprotein E	−5.889	1.47 × 10^−11^	9.71 × 10^−8^
	TSPAN12	Tetraspanin 12	−4.435	3.64 × 10^−7^	0.001
	PRSS35	Protease, serine 35	−4.249	2.21 × 10^−6^	0.004
	ASS1	Argininosuccinate Synthase 1	−3.786	6.89 × 10^−7^	0.002
	SULF1	Sulfatase 1	−3.338	3.63 × 10^−5^	0.028
Upregulated	METRN	Meteorin, glial cell differentiation regulator	2.872	5.07 × 10^−6^	0.007
	PRAP1	Proline rich acidic Protein 1	2.852	2.85 × 10^−5^	0.024
	RAMP1	Receptor activity-modifying protein 1	2.311	1.56 × 10^−5^	0.016
	TMEM54	Transmembrane protein 54	1.921	1.37 × 10^−6^	0.003

## Data Availability

The data that support the findings of this study are available from the corresponding author, Y.L., upon reasonable request.

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
