# Peer review of "Transcriptomics Analysis of Lens from Patients with Posterior Subcapsular Congenital Cataract"

_genes, 2021, doi:10.3390/genes12121904_

Round 1
Reviewer 1 Report
This is an interesting work that investigates by transcriptomics analyses on the differentially expressed genes in patients with posterior subcapsular congenital cataract, collecting information, among others, on such genes and the biological pathways most significatively affected. It looks correctly made at experimental and transcriptional data analysis, and the results could have therapeutic value for the prevention and treatment of the disease. In this reviewer's opinion, the manuscript would benefit of revision on distinct aspects, including:
1.-At the abstract, the work mentions the identification of 2,300 and 3,009 differentially expressed genes among 55,099 and 46,176 genes, respectively, in the epithelial cells and in lens cortex/nuclear fibres of such sub capsular congenital cataract location. It would be advisable that such description is improved, i.e. clarifying the "gene term" used here (protein-coding genes? or protein-coding + non-coding genes?), given that nowadays is widely accepted that in human genomes the number of protein-coding genes is much below such 55,099/ 46,176 numbers. Also noteworthy that, at the abstract, the abbreviation DEGs appears at line 20, but it is not defined until line 112 at Materials and Methods.
2.-At paragraph of lines 293-299 the workflow analysis used in the work is explained and is said that it has been suggested by Anand D et al, but it is not clear enough whether this is actually a direct suggestion from such researcher and colleagues or it just follows the procedure described by them in refs. 12, 13, 14, and 5 commented along this paragraph. In the next paragraph, at lines 306-307, it is said that, among others, the transcriptomic analyses indicate that after cataract formation distinct biological pathway are altered, like lens cell development, cell differentiation and growth, as well as metal ions effects, like cadmium, but this last effect is not well explained in such sentence. Both paragraphs should be revised accordingly.
3.-The paragraph between lines 346-374 deals with HTRA1, a secreted serine protease that has been found in the work that its gene is one of the most significatively altered DEGs in posterior congenital subcapsular cataract patients. In several parts of such paragraph it is said that such protein binds or interacts with TGF-beta family proteins, inhibiting their functions, and authors assume that may modulate progression of the studied cataract by interfering with the TGF-beta signalling. However, in such hypothesis its is not mentioned or discussed that HTRA1 is a protease, with clear proteolytic capability and recognised effect in cartilage degeneration, skeletal disorders and cancer (Y Li et al, 2020, Cell Prolif) and, therefore, that such interaction and/or modulation/regulation could take place through proteolysis. It would be wise to indicate and substantiate it.
4.-At references 14 and 19, the way to indicate the "doi" number probably is not correct and have to be mended.
Author Response
Manuscript ID: genes-1449240
Title: Transcriptomics Analysis of Lens from Patients with Posterior Subcapsular Congenital Cataract
Dear Editors and Reviewers:Thank you very much for your letter and advice. We have revised the manuscript accordingly and would like to resubmit it for your consideration. We are very grateful for the reviewers’ comments and suggestions, and we hope that we have addressed each of them adequately. If you have any questions, please do not hesitate to contact me at the email address below.
Yours sincerely,
Yi Luo
Corresponding author
E-mail: [email protected]
Reviewer #1:
Thank you for your useful comments. Below are our responses to each of your suggestions.
Comment 1:
This is an interesting work that investigates by transcriptomics analyses on the differentially expressed genes in patients with posterior subcapsular congenital cataract, collecting information, among others, on such genes and the biological pathways most significatively affected. It looks correctly made at experimental and transcriptional data analysis, and the results could have therapeutic value for the prevention and treatment of the disease. In this reviewer's opinion, the manuscript would benefit of revision on distinct aspects, including:
1.At the abstract, the work mentions the identification of 2,300 and 3,009 differentially expressed genes among 55,099 and 46,176 genes, respectively, in the epithelial cells and in lens cortex/nuclear fibres of such sub capsular congenital cataract location. It would be advisable that such description is improved, i.e. clarifying the "gene term" used here (protein-coding genes? or protein-coding + non-coding genes?), given that nowadays is widely accepted that in human genomes the number of protein-coding genes is much below such 55,099/ 46,176 numbers. Also noteworthy that, at the abstract, the abbreviation DEGs appears at line 20, but it is not defined until line 112 at Materials and Methods.
Response:
Thank you for affirming the usefulness of our work and scrutinising our manuscript. We have added the definition of the abbreviation DEGs (differentially expressed genes) in the abstract, and we have carefully checked the results of the mRNA-sequencing. We used the Ensembl gene annotation system, and the 55,099 and 46,176 genes identified in this study include all the genes identified with Ensembl IDs. Most of the genes were protein-coding genes and a small number of non-coding genes were also identified. As the library for mRNA-sequencing is poly(A)+ enriched, some non-coding genes with poly(A) tail were also identified. And the results also included predictive genes, unknown functional genes, and pseudogenes with no gene description. We have deleted these genes and rephrased the abstract and results accordingly. We have also updated Figure 1 using the new results. It now reads in the abstract as follows:
“There were 1,533 differentially expressed genes from the 19,072 genes in lens epithelial cells of posterior subcapsular congenital cataract patients compared to normal controls at a cut-off criterion of |log2 Fold Change| > 1 and P < 0.05, including 847 downregulated genes and 686 upregulated genes. To further narrow down the DEGs, we utilised the stricter criteria of |log2 Fold Change| > 1 and an FDR value < 0.05, and we identified 551 DEGs, including 97 upregulated genes and 454 downregulated genes. This study also identified 1,263 differentially expressed genes of the 18,755 genes in lens cortex and nuclear fibres, including 646 downregulated genes and 617 upregulated genes.”
Comment 2:
At paragraph of lines 293-299 the workflow analysis used in the work is explained and is said that it has been suggested by Anand D et al, but it is not clear enough whether this is actually a direct suggestion from such researcher and colleagues or it just follows the procedure described by them in refs. 12, 13, 14, and 5 commented along this paragraph. In the next paragraph, at lines 306-307, it is said that, among others, the transcriptomic analyses indicate that after cataract formation distinct biological pathway are altered, like lens cell development, cell differentiation and growth, as well as metal ions effects, like cadmium, but this last effect is not well explained in such sentence. Both paragraphs should be revised accordingly.
Response:
Thank you for your kind advice. We have examined the suggested article. Previous studies have used the iSyTE database to identify new genes associated with cataracts [1-3], and Anand D et al. commented that the enrichment strategy of the iSyTE database is advantageous because it identifies transcripts that might not necessarily have high absolute expression yet have high enrichment in a tissue [4]. The paragraph including lines 293-299 has been rephrased. It now reads as follows:
“The iSyTE database allows visualisation and analysis of dynamic gene expression in lens development and disease, helps to identify new cataract-linked genes [5], and provides an estimate of the expression of lens enrichment. Previous studies have used the database to identify new genes associated with cataracts [5,12,13,14].”
In the paragraph including lines 306-307, we have explained the metal ions' effects on cataracts. It now reads as follows:
“The functional cluster analysis of the identified genes showed that a variety of biological pathways changed significantly after posterior subcapsular congenital cataract formation, including lens cell development, cell differentiation and growth, and cadmium ion. Choudhary et al. found that six weeks of CdCl2 administration resulted in considerable progress of cataract formation, and the antioxidants for lenses, such as CAT, SOD, GPx and GSH, were significantly reduced [16]. Heavy metal ions might play a potential role in cataract pathology [15-17].”
Comment 3:
The paragraph between lines 346-374 deals with HTRA1, a secreted serine protease that has been found in the work that its gene is one of the most significatively altered DEGs in posterior congenital subcapsular cataract patients. In several parts of such paragraph it is said that such protein binds or interacts with TGF-beta family proteins, inhibiting their functions, and authors assume that may modulate progression of the studied cataract by interfering with the TGF signalling. However, in such hypothesis its is not mentioned or discussed that HTRA1 is a protease, with clear proteolytic capability and recognised effect in cartilage degeneration, skeletal disorders and cancer (Y Li et al, 2020, Cell Prolif) and, therefore, that such interaction and/or modulation/regulation could take place through proteolysis. It would be wise to indicate and substantiate it.
Response:
We agree with the reviewer and sincerely appreciate the comment. We have added a clear explanation of the proteolytic capability of HTRA1 and the interaction between HTRA1 and TGF-β. The paragraph has been rephrased. It now reads as follows:
“It is reported that HTRA1 interacts with members of the TGF-β family and regulates their signalling pathway [31]. The GST pull-down analysis showed that HTRA1 protein bound to all tested TGF-β family proteins and inhibited their function [32]. The interaction between HTRA1 and the TGF-β family has been verified in various diseases, including brain development [33], age-related macular degeneration [34,35], and tumours [36]. As a serine protease, HTRA1 plays a key role in regulating various cellular processes and participates in cell proliferation, migration and apoptosis. Previous studies have found that HTRA1 has a clear proteolytic capability and a recognised effect in cartilage degeneration, skeletal disorders and cancers [36], and HTRA1 can bind to TGF-β1 and degrade and regulate the amount of TGF-β1 in vivo and in vitro [31, 37].”
Comment 4:
At references 14 and 19, the way to indicate the "doi" number probably is not correct and have to be mended.
Response:
We appreciate the suggestion of the reviewer. The "doi" numbers have been mended.
END OF COMMENTS
References:
1.Agrawal SA, Anand D, Siddam AD, et al. Compound mouse mutants of bZIP transcription factors Mafg and Mafk reveal a regulatory network of non-crystallin genes associated with cataract. Hum Genet 2015;134(7):717-35.
2.Anand D, Agrawal SA, Siddam AD, et al. An integrative approach to analyze microarray datasets for prioritization of genes relevant to lens biology and disease. Genomics Data 2015;5:223-7.
3.Kakrana A, Yang A, Anand D, et al. iSyTE 2.0: a database for expression-based gene discovery in the eye. Nucleic Acids Res 2018;46(D1):D875-85.
4.Anand D, Lachke SA. Systems biology of lens development: A paradigm for disease gene discovery in the eye. Exp Eye Res 2017;156:22-33.

Reviewer 2 Report
These authors have attempted to address an important problem: What are the differences in gene expression between normal lenses and ones with congenital posterior capsular cataracts. This will be a useful addition to the literature. However, the impact of this paper is somewhat limited, because there are no discoveries of new cataract genes or mechanistic additions to our understanding of cataractogenesis.
I also have a few specific comments:
- There seems to be an error in the abstract. It reads, “More than 2,300 of the 55,009 genes in lens epithelial cells of posterior subcapsular congenital cataract patients differ more than twofold compared to normal controls. Of these, 122 genes are upregulated and 531 genes are downregulated.” I do not understand: 122 + 531 < 2300. ???
- I would like to see more information regarding the quality of the RNA isolated from these patients. It is not easy to isolate high quality (non-degraded) mRNA from lenses (especially cortex and nuclear fibers).
- One of the most striking findings is the big differences in expression of different hemoglobin chains. The authors need to further investigate this and to discuss it more thoroughly. One trivial explanation (that must be addressed or excluded) is that the epithelial cell samples of the cataract patients were contaminated with reticulocytes (or perhaps the cataracts were partially vascularized or perhaps there were more remnants of the fetal vasculature.
Author Response
Manuscript ID: genes-1449240
Title: Transcriptomics Analysis of Lens from Patients with Posterior Subcapsular Congenital Cataract
Dear Editors and Reviewers:Thank you very much for your letter and advice. We have revised the manuscript accordingly and would like to resubmit it for your consideration. We are very grateful for the reviewers’ comments and suggestions, and we hope that we have addressed each of them adequately. If you have any questions, please do not hesitate to contact me at the email address below.
Yours sincerely,
Yi Luo
Corresponding author
E-mail: [email protected]
Reviewer #2:
Thank you very much for your useful comments. The below is our responses with each of your suggestions.
Comment 1:
These authors have attempted to address an important problem: What are the differences in gene expression between normal lenses and ones with congenital posterior capsular cataracts. This will be a useful addition to the literature. However, the impact of this paper is somewhat limited, because there are no discoveries of new cataract genes or mechanistic additions to our understanding of cataractogenesis.
Response:
We appreciate that you value our work and provide this useful comment. To identify the candidate genes from the differentially expressed genes (DEGs) obtained via RNA-seq analysis that might be associated with the pathology of posterior congenital cataract, we analysed the DEGs using the iSyTE2.0 database resource [1], which represented the gene expression and lens enrichment data during normal lens development. Previous studies have suggested that the iSyTE database can serve as a cataract gene prioritisation resource [2], and that 95% of the known genes associated with cataracts or lens defects are lens-enriched at one or more key embryonic stages [3]. After filtering the DEGs by iSyTE2.0, the top lens-enriched genes at key embryonic stages were obtained. These genes were filtered by the Cat-Map database to exclude the genes known to be associated with cataracts, and the remaining genes became the high-priority candidate genes for cataractogenesis, such as GRIFIN, DAPL1, LGALS3, GPR160, MBOAT1, HTRA1, SLC7A2, PIP5KL1, CAPRIN2 and ARSI. These genes were enriched in the Wnt signalling pathway, apoptotic signalling pathway and TGF-β signalling pathway. The results could provide a new direction for the study of congenital cataract. However, the function and mechanism of high-priority candidate genes are still unknown and should be verified in further studies. Further studies are in progress in our research group.
Comment 2:
- There seems to be an error in the abstract. It reads, “More than 2,300 of the 55,009 genes in lens epithelial cells of posterior subcapsular congenital cataract patients differ more than twofold compared to normal controls. Of these, 122 genes are upregulated and 531 genes are downregulated.” I do not understand: 122 + 531 < 2300. ???
Response:
We appreciate this comment and sincerely apologise for the error. The DEGs in the lens epithelial cells between the two groups were analysed using two cut-off criteria. Using the cut-off criteria of |log2 Fold Change| > 1 and P < 0.05, we identified 2,376 DEGs of the 55,009 genes in lens epithelial cells, including 1132 downregulated genes and 1244 upregulated genes. In order to decrease the number of false positives among significant results, we further used a stricter standard--- false discovery rate (FDR) value as a cut-off criterion. Using the cut-off criteria of |log2 Fold Change| > 1 and an FDR value < 0.05, we identified 653 DEGs, including 122 upregulated genes and 531 downregulated genes.
We have carefully checked the results of the mRNA-sequencing and found the total number of identified genes need to be rephrased. We used the Ensembl gene annotation system, and the 55,099 and 46,176 genes identified in this study include all the genes identified with Ensembl IDs. Most of the genes were protein-coding genes and a small number of non-coding genes were also identified. As the library for mRNA-sequencing is poly(A)+ enriched, some non-coding genes with poly(A) tail were also identified. And the results also included predictive genes, unknown functional genes, and pseudogenes with no gene description. We have deleted these genes and rephrased the abstract and results accordingly. We have also updated Figure 1 using the new results. It now reads in the abstract as follows:
“There were 1,533 differentially expressed genes from the 19,072 genes in lens epithelial cells of posterior subcapsular congenital cataract patients compared to normal controls at a cut-off criterion of |log2 Fold Change| > 1 and P < 0.05, including 847 downregulated genes and 686 upregulated genes. To further narrow down the DEGs, we utilised the stricter criteria of |log2 Fold Change| > 1 and an FDR value < 0.05, and we identified 551 DEGs, including 97 upregulated genes and 454 downregulated genes. This study also identified 1,263 differentially expressed genes of the 18,755 genes in lens cortex and nuclear fibres, including 646 downregulated genes and 617 upregulated genes.”
Comment 3:
I would like to see more information regarding the quality of the RNA isolated from these patients. It is not easy to isolate high quality (non-degraded) mRNA from lenses (especially cortex and nuclear fibers).
Response:
Thank you for your kind advice. The concentration of total RNA was measured with Qubit 4.0 (Thermo, USA). As shown in the figure below(please see the attachment), the OD260/OD280 ratios were between 1.8 and 2.0, and the curves were acceptable. The total RNA was higher than 200 ng for all samples included in this study. Figures 1A and 1B include sample reports from anterior lens capsules. Figures 1C and 1D include sample reports from the lens cortex and nuclear fibers.
Figure 1. Reports of the concentration of total RNA measured by Qubit 4.0.
The RNA quality was also analysed using agarose gel electrophoresis. As shown in the figure below (please see the attachment), the C lane is the standard sample and lanes 5,6, and 7 are the RNA samples from lens cortex and nuclear fibers. The results showed that the 18S and 28S ribosomal RNA bands were clearly visible in the RNA samples. The RNA samples were intact and not degraded.
Figure 2. Agarose gel electrophoresis for RNA samples from lens cortex and nuclear fibers.
After reverse transcription, the cDNA quality was analysed using the Agilent 2100 Bioanalyzer (Agilent Technologies, CA, USA). Figures 3A and 3B (please see the attachment) are images from the anterior lens capsules. Figures 3C and 3D are images from the lens cortex and nuclear fibers. The ratios of the fragments higher than 650 bp were higher than 70% for all samples included in the study.
Figure 3. Agilent 2100 Bioanalyzer analysis of cDNA extracted from anterior lens capsules and lens cortex and nuclear fibers.
For the RT-qPCR samples, we checked the amplification plot and melt curve for every sample. The figure below (please see the attachment) is an example of the amplification plot and melt curve of the RT-qPCR from samples of the lens cortex and nuclear fibers.
Figure 4. The amplification plot and melt curve of the RT-qPCR for samples from lens cortex and nuclear fibers.
Comment 4:
One of the most striking findings is the big differences in expression of different hemoglobin chains. The authors need to further investigate this and to discuss it more thoroughly. One trivial explanation (that must be addressed or excluded) is that the epithelial cell samples of the cataract patients were contaminated with reticulocytes (or perhaps the cataracts were partially vascularized or perhaps there were more remnants of the fetal vasculature.
Response:
We agree with the reviewer and sincerely appreciate the comment. We have tried to investigate the reason for the increased expression of different haemoglobin chains. Patients with partially vascularised cataract, persistent fetal vasculature (PFV) or persistent hyperplastic primary vitreous (PHPV) were not included in this study. The inclusion and exclusion criteria have been added to the section of “Subjects”. We have also carefully replayed the surgical videos for the congenital cataract patients. During the procedure to make a scleral tunnel incision, a small amount of bleeding at the incision cannot be avoided because of the rich blood vessels at the site. The sclera electrocoagulation is effective to control the bleeding, but it can also significantly increase local inflammation, delay the wound healing and increase the risk of endophthalmitis, which is one of the most serious complications after surgery. Therefore, we did not use sclera electrocoagulation during the surgery. While we have tried to use gauze and flowing phosphate buffered saline (PBS) to clean the blood at the incision before the epithelial cell sample was taken out from it. And immediately after the samples were taken out, we used PBS to clean the samples for at least three times to avoid the contamination by reticulocytes. At the beginning of the study, we tried using the red blood cell lysate to remove the rest of the red blood cells, but after the lysis, the number of lens epithelial cells also decreased significantly, and we were concerned about the impact on the results of RNA-sequencing. Therefore, red blood cell lysate was not used in this study. The increased expression of different haemoglobin chains in the epithelial cell samples cannot excluded the probability of residual red blood cells. Thanks again for your kind advice. We will fully consider this potential influencing factor in our further study and expand sample size to verify the candidate genes. We have further explained these results in the discussion. It now reads as follows:
“Interestingly, the results show that there are 5 significantly upregulated genes, including HBA2, HBA1, HBB, HBG1, and HBG2, which encode different haemoglobin chains and are enriched in haemoglobin complex, oxygen carrier activity and oxygen transport. The reason for these results cannot excluded the probability of residual red blood cells during the surgery. Although we have used phosphate buffered saline (PBS) to clean the samples for at least three times to avoid the remaining reticulocytes after the samples were taken out from the incision, a small number of red blood cells may remain in the sample. More samples should be enrolled to verify the function and mechanism of high-priority candidate genes in further studies.”
END OF COMMENTS
References:
1.Kakrana A, Yang A, Anand D, et al. iSyTE 2.0: a database for expression-based gene discovery in the eye. Nucleic Acids Res 2018;46(D1):D875-85.
2.Liu H, Barnes J, Pedrosa E, et al. Transcriptome analysis of neural progenitor cells derived from Lowe syndrome induced pluripotent stem cells: identification of candidate genes for the neurodevelopmental and eye manifestations. J Neurodev Disord 2020;12(1):14.
3.Anand D, Kakrana A, Siddam AD, et al. RNA sequencing-based transcriptomic profiles of embryonic lens development for cataract gene discovery. Hum Genet 2018;137(11-12):941-54.
Reviewer 3 Report
The authors presented an interesting study comparing the mRNA expression in children with posterior subcapsular congenital cataract compared to normal children. The manuscript is with merit and the findings are worth reporting. However,the authors should revise the manuscript and address the following concerns.
- 1 Subjects: Line 64: correct the repetition “2.1. 2.1. Subjects”
- Participants: a list of inclusion and exclusion criteria should be provided.
- The information about the details regarding the type of cataract in the enrolled subjects and their mean age should be moved from this section “2.1 Subjects” (lines 70-71, 79-80) to the beginning of the “Results” section.
- The sentence “The age did not differ significantly between the congenital cataract patients and normal controls (P>0.05)” (line 82-83) should be moved from this section “2.1 Subjects” to the “Results” section.
- Statistics: The author should provide a statistical power estimation for their study or at least some justification of the study n and add it in a separate section of the methods entitled “Statistical analysis”
- The authors should expand their conclusions sections and provide some additional insights about the potential clinical relevance of their research in terms of disease prevention and treatment
Author Response
Manuscript ID: genes-1449240
Title: Transcriptomics Analysis of Lens from Patients with Posterior Subcapsular Congenital Cataract
Dear Editors and Reviewers:Thank you very much for your letter and advice. We have revised the manuscript accordingly and would like to resubmit it for your consideration. We are very grateful for the reviewers’ comments and suggestions, and we hope that we have addressed each of them adequately. If you have any questions, please do not hesitate to contact me at the email address below.
Yours sincerely,
Yi Luo
Corresponding author
E-mail: [email protected]
Reviewer #3:
Thank you very much for your useful comments. Below are our responses to each of your suggestions.
Comment 1:
The authors presented an interesting study comparing the mRNA expression in children with posterior subcapsular congenital cataract compared to normal children. The manuscript is with merit and the findings are worth reporting. However, the authors should revise the manuscript and address the following concerns.
Subjects: Line 64: correct the repetition “2.1. 2.1. Subjects”
Response:
Thank you for your affirmation of the usefulness of our work and this useful comment. The repetition has been corrected.
Comment 2:
Participants: a list of inclusion and exclusion criteria should be provided.
Response:
We agree with the reviewer and sincerely appreciate the comment. The inclusion and exclusion criteria have been added to the section of “Subjects”. It now reads as follows:
“Patients with bilateral or unilateral posterior subcapsular congenital cataracts under 10 years of age were included in the study. For children with posterior subcapsular congenital cataract, the exclusion criteria were as follows: ocular inflammation, persistent fetal vasculature (PFV), persistent hyperplastic primary vitreous (PHPV), pre-existing posterior capsule defect, combined other developmental disorder, or a history of ocular surgery or trauma. The normal controls included transparent lenses from children younger than 10 years of age.”
Comment 3:
The information about the details regarding the type of cataract in the enrolled subjects and their mean age should be moved from this section “2.1 Subjects” (lines 70-71, 79-80) to the beginning of the “Results” section.
Response:
Thank you very much for your kind advice. We have moved the detailed information to the “Results” section as suggested.
Comment 4:
The sentence “The age did not differ significantly between the congenital cataract patients and normal controls (P>0.05)” (line 82-83) should be moved from this section “2.1 Subjects” to the “Results” section.
Response:
We appreciate the comment of the reviewer and the sentence have moved to the “Results” section.
Comment 5:
Statistics: The author should provide a statistical power estimation for their study or at least some justification of the study n and add it in a separate section of the methods entitled “Statistical analysis”
Response:
We appreciate the reviewer’s suggestion. The statistical power estimation was also an issue we were concerned about. The replication number and the choice for sequencing mainly determine the power [1]. In this study, we used deep sequencing, and an average of 110,375,304 (79,150,688–167,149,114) raw reads were yielded per sample. Previous studies have suggested that higher sequencing depth generates more informational reads, which increases the statistical power to detect differentially expressed genes [1, 2]. For the replication number, usually at least 3 biological replications are required in every group, and increasing the number of biological replications increases the power significantly [1]. Because of the relatively low incidence of congenital cataracts and the lack of voluntary child donors, the sample size was limited and could not fully represent the expression profiles of all patients. In future studies, we will recruit more patients with congenital cataracts and various cataract phenotypes to determine the expression differences among them. We have added these limitations to the end of the discussion.
In addition, we have added a “Statistical analysis” subsection to the Methods section. It now reads as follows:
“Differential expression statistics were assessed using DESeq2, and the adjusted P-values which was the false discovery rate (FDR) were obtained using the Benjamini–Hochberg multiple test correction method. The genes with |log2 Fold Change| > 1 and a P value < 0.05 were defined as a DEG between the two groups. The GO terms and KEGG pathway with P values less than 0.05 were considered to be significantly enriched. Statistical analyses for age and RT-qPCR results were performed using SPSS 22.0 (IBM, Armonk, NY). The Kolmogorov-Smirnov test was used for normality distribution. An independent sample t test or Mann-Whitney U test was performed to analyse the difference between the posterior subcapsular congenital cataract patients and the normal control group. P values less than 0.05 were considered statistically significant.”
Comment 6:
The authors should expand their conclusions sections and provide some additional insights about the potential clinical relevance of their research in terms of disease prevention and treatment
Response:
We agree with the reviewer. We have rephrased the Conclusions section. It now reads as follows:
“The present study provides evidence of broad differences in gene expression between the lenses in children with posterior subcapsular congenital cataract and the lenses in normal children, and identify novel candidate genes. The gene functional enrichment analysis might deepen our understanding of posterior subcapsular congenital cataract formation and mechanisms. Furthermore, this study provides promising evidence that can inform future research on the pathological mechanisms of posterior subcapsular congenital cataracts and the screening of clinical candidate drugs. These investigations could help prevent and treat this idiopathic cataract.”
END OF COMMENTS
References:
- Liu Y, Zhou J, White KP. RNA-seq differential expression studies: more sequence or more replication? Bioinformatics 2014;30(3):301-4.
- Tarazona, S.; García-Alcalde, F.; Dopazo, J.; Ferrer, A.; Conesa, A. Differential expression in RNA-seq: a matter of depth. Genome Res. 2011, 21, 2213-2223.
